# Persistence with Botulinum Toxin Treatment for Spasticity Symptoms in Multiple Sclerosis

**DOI:** 10.3390/toxins14110774

**Published:** 2022-11-09

**Authors:** Federica Novarella, Antonio Carotenuto, Paolo Cipullo, Rosa Iodice, Emanuele Cassano, Antonio Luca Spiezia, Nicola Capasso, Maria Petracca, Fabrizia Falco, Carmine Iacovazzo, Giuseppe Servillo, Roberta Lanzillo, Vincenzo Brescia Morra, Marcello Moccia

**Affiliations:** 1Multiple Sclerosis Clinical Care Unit, Federico II University Hospital, 80131 Naples, Italy; 2Department of Neurosciences, Federico II University of Naples, 80131 Naples, Italy; 3Department of Human Neurosciences, Sapienza University of Rome, 00185 Rome, Italy; 4Department of Molecular Medicine and Medical Biotechnology, Federico II University of Naples, 80131 Naples, Italy

**Keywords:** multiple sclerosis, botulinum, spasticity, persistence

## Abstract

Botulinum toxin (BT) is an effective treatment for spasticity symptoms in multiple sclerosis (MS). Despite its wide use in clinical practices, only few studies have explored long-term persistence. We aim to evaluate the rate of discontinuation of BT treatment and the correlation with MS, spasticity, and injection variables. This retrospective study on 3-year prospectively collected data included 122 MS patients receiving BT injections for spasticity. We collected MS clinical variables (disease durations, Expanded Disability Status Scales [EDSSs], disease-modifying treatments [DMT], and Symbol Digit Modalities Tests [SDMTs]), modified Ashworth scales [MASs], concomitant treatments, and injection variables (formulation, dose, number of injections, and intervals between injections). A total of 14 out of the 122 patients discontinued BT after a mean time of 3.0 ± 1.5 years. In the Cox regression model including the MS clinical variables, the probability of BT discontinuations increased in patients with DMT changes during follow-ups (HR = 6.34; 95%Cl = 2.47, 18.08; *p* < 0.01) and with impaired SDMTs (HR = 1.20; 95%Cl = 1.04, 1.96; *p* < 0.01). In the model including the spasticity variables, there were no associations between BT discontinuation and MAS or other spasticity treatments. In the model including the injection variables, the probability of discontinuation decreased by 80% for each cumulative injection (HR = 0.16; 95%Cl = 0.05, 0.45; *p* < 0.01), but increased by 1% for each additional day over the 3-month interval between injections (HR = 1.27; 95%Cl = 1.07, 1.83; *p* < 0.01). BT discontinuation was associated with concomitant MS-related issues (e.g., treatment failure and DMT change) and the presence of cognitive impairment, which should be accounted for when planning injections. The interval between injections should be kept as short as possible from regulatory and clinical perspectives to maximize the response across all of the spasticity symptoms and to reduce discontinuation in the long term.

## 1. Introduction

Spasticity is a clinically meaningful condition in over 80% of people with multiple sclerosis (MS), and significantly affects quality of life [1,2]. Botulinum toxin (BT) injections safely and effectively treat spasticity by temporarily blocking neuromuscular junctions, thus, favoring the patients’ and caregivers’ possibilities to exploit residual function and improve the outcome of rehabilitation treatments [3,4,5,6,7]. Clinical trials are generally run in a limited time frame (i.e., response at 4–12 weeks) and include a variety of neurological conditions, thus, limiting the analysis of disease-specific features. Similarly, real-world studies have focused on short-term improvements following the BT injections (e.g., up to 12 months) [5], while long-term use has been poorly investigated [8,9].

All MS patients with spasticity would potentially benefit from BT injections; thus, it would be useful to know the factors that can predict therapeutic responses to customize treatments. In the present study, we used persistence with BT as a marker of treatment efficacy and safety [10,11,12], and we aimed to evaluate the risk of discontinuation in relation to MS clinical, spasticity, and injection-related features. We hypothesize that there are individual factors increasing the risks of discontinuation, which should, therefore, be considered in clinical practice for customized treatment decisions.

## 2. Results

We included 122 patients with MS receiving BT injections for spasticity symptoms during the study period, among whom 14 discontinued the treatment after 3 years of follow-ups (11.4%). The demographic, MS, spasticity, and injection variables are reported in Table 1. The MS patients who discontinued the BT injections were younger (*p* = 0.02) and had longer intervals between the injections (*p* < 0.01). During the follow-ups, 20 MS patients had EDSS progression, and 2 patients had a clinical relapse. No serious adverse events were recorded. The most common injection goal was posturing/hygiene (*n* = 40, 32.8%), followed by mobility (*n* = 35, 28.7%), pain (*n* = 34, 27.9%), and daily assistance/functioning in daily living activities (*n* = 13, 10.6%). The concomitant spasticity treatments included cannabinoids (*n* = 31), baclofen (*n* = 20), benzodiazepines (*n* = 8), gabapentin (*n* = 6), 4-aminopiridine (*n* = 5), pregabalin (*n* = 2), and tizanidine (*n* = 1).

In the Cox regression model including the MS variables, the probability of BT discontinuation increased by six-fold in patients with DMT changes during follow-up (HR = 6.34; 95%CI = 2.47, 18.08; *p* < 0.01) (Figure 1a) and by 20% in patients with impaired SDMT (HR = 1.20; 95%CI = 1.04, 1.96; *p* < 0.01) (Figure 1b). We found no significant association for the disease duration (HR = 1.03; 95%CI = 0.95, 1.12; *p* = 0.45), EDSS (HR = 0.81; 95%CI = 0.54, 1.20; *p* = 0.30), or DMT efficacy at the first injection (HR = 0.58; 95%CI = 0.12, 2.77; *p* = 0.50).

In the Cox regression model including the spasticity variables, we found no significant association between the probability of BT discontinuation and the highest MAS score (HR = 1.14; 95%CI = 0.38, 3.39; *p* = 0.81) or the concomitant spasticity treatments (HR = 1.68; 95%CI = 0.53, 5.25; *p* = 0.37).

In the Cox regression model including the injection variables, the probability of BT discontinuation decreased by 80% for each cumulative injection (HR = 0.16; 95%CI = 0.05, 0.45; *p* < 0.01) (Figure 1c) and increased by 1% for each additional interval day between the injections compared with the conventional 3-month interval (HR = 1.27; 95%CI = 1.07, 1.83; *p* < 0.01) (Figure 1d). We found no significant association for the BT formulation at the first injection (abobotulinumtoxin A HR = 0.90; 95%CI = 0.08, 3.80; *p* = 0.93; incobotulinumtoxin A HR = 8.30; 95%CI = 0.20, 34.79; *p* = 0.49; using onabotulinumtoxin A as a reference), the BT dose at the first injection (HR = 0.99; 95%CI = 0.99, 1.01; *p* = 0.35), or changes in the BT preparation or dosing (HR = 0.74; 95%CI = 0.08, 6.85; *p* = 0.79).

Finally, in the stepwise Cox regression model including all of the variables, we confirmed the associations for the cumulative number of injections (HR = 0.17; 95%CI = 0.06, 0.46; *p* < 0.01) and the intervals between the injections (HR = 3.15; 95%CI = 2.10, 4.71; *p* < 0.01).

## 3. Discussion

We evaluated the long-term rate of the BT treatment persistence in relation to MS clinical, spasticity, and injection-related characteristics. In particular, we showed an overall high rate of persistence with BT treatment, which decreased in patients with a DMT change, an impaired SDMT, a lower number of injections, and longer intervals between the injections. As such, we have identified some factors, potentially amenable to intervention, to be accounted for in clinical practice while defining treatment strategies for spasticity symptoms. A strength of our study is that it includes a large and homogeneous sample of MS patients who are treated with BT exclusively for spasticity symptoms, with a 3-year follow-up.

Overall, we found that almost 90% of the patients remained on treatment with BT during the follow-ups, which is significantly higher than a previous similar study (i.e., 56% discontinuation after 1.2 years) [12]. This could be due to the presence of a BT clinic that is specifically dedicated to MS spasticity symptoms. Additionally, in line with previous studies [3], the most common injection target was posturing/hygiene, followed by mobility, pain, and daily assistance/functioning in daily living activities. The injection goal did not affect the persistence with BT treatment, suggesting that all of the goals can be equally achieved with good efficacy and tolerability. Furthermore, among the spasticity variables, neither the MAS score nor the concomitant treatments for spasticity were associated with BT discontinuation, thus, confirming the effect of BT alone or in combination with other treatments for any severity of the spasticity (the MAS was between 1 and 4 in our study) [13]. Finally, looking at the injection variables, no association was found between the BT discontinuation and the use of different BT formulations or doses, thus, suggesting that similar efficacy can be achieved.

The rate of BT discontinuation was associated with DMT changes. This result, rather than demonstrating the direct effect of DMTs on the response to BT, suggests that the patients’ and physicians’ efforts were aimed at controlling the disease activity. In the future, an integrated approach between the different care needs should be considered to ensure greater persistence with BT. In keeping with this, we showed that the patients with the highest number of injections stayed longer on the BT treatment, suggesting that clinical efficacy was achieved progressively during the therapeutic plan with BT and that it was then maintained over time.

The rate of BT discontinuation was higher in patients with cognitive impairment, as measured with the SDMT, supporting the association between cognitive function and their ability to comply with the treatment. A lower score on the SDMT is already recognized as a risk factor for worse disease progression and for worse overall functioning in daily living activities [14]. Our work, therefore, confirms the importance of cognitive impairment in identifying a subgroup of particularly vulnerable MS patients and even in the possibility of benefiting from symptomatic therapy. In particular, the support of a caregiver could overcome this limitation [12].

Our study also suggests that the interval between the injections must be kept as short as possible (i.e., ≤3.25 months), considering a 1% increase in the discontinuation rate for each additional average interval day. Intriguingly, we found a similar follow-up duration in patients with and without BT discontinuation that, on the contrary, differed in their total number of injections and intervals between injections. This finding further suggests that short infusion intervals maximize the BT response over a similar follow-up. Short infusion intervals have been previously suggested to maximize the treatment response to BT [15]; they should be considered within an individual approach program to manage spasticity and related symptoms. 

Several limitations may have affected our results, including the recruitment from a single Italian MS Clinical Care Unit and the difficulties in generalizing our findings to the MS population. Similarly, we included a limited sample of patients with MS, while our evidence should be confirmed in a larger population, including patients with spasticity from other etiologies. The time interval of the follow-ups could have been longer, and other possible associations could have been sought, including different clinical subtypes of MS. We did not analyze target muscles but preferred injection goals due to sample size constraints and the willingness to evaluate the actual benefits in daily life activities [16]. Additionally, we collected only serious adverse events, while a more in-depth investigation could have led to other statistical associations and, possibly, improved persistence [17]. We did not collect reasons for discontinuation, which can include logistics, switches to other spasticity treatments, poor efficacy, and/or tolerability [18]. However, looking at our results, disease worsening in terms of cognitive impairment and DMT failure are most likely responsible for BT discontinuation.

## 4. Conclusions

Based on the MS clinical, spasticity, and injection variables, we showed that in order to maintain patients with MS on treatment with BT for spasticity symptoms (1) the interval between injections must be kept as short as possible from a regulatory and clinical point of view and (2) other MS-related issues (e.g., DMT change and cognitive impairment) should be accounted for within customized treatment strategies and integrated approaches. Overall, MS and BT specialists should share an appropriate treatment plan with MS patients, and include a combination of spasticity, BT injections, and MS variables to foster long-term persistence.

## 5. Methods

### 5.1. Study Design and Population

This is a retrospective study on 3-year prospectively collected data from 122 patients with MS receiving BT injections for spasticity at the MS Clinical Care Unit, Federico II University Hospital, Naples, Italy (the database was completed on April 2022).

Ethics approval was obtained from the committee of Federico II University of Naples, Italy (355/19). All of the patients signed informed consents prior to the study. The study included anonymized data collected in clinical practice (GDPR 2016/679), and it was conducted in accordance with good clinical practice and the Declaration of Helsinki.

The inclusion criteria were as follows: (1) a diagnosis of MS [19]; (2) the presence of at least 3 injections of BT for MS-related spasticity; (3) clinical stability (no clinical relapses nor DMT changes in the 3 months before the first BT injection). The exclusion criterion was as follows: (1) incomplete medical records.

### 5.2. Demographics and MS-Related Clinical Variables

At the time of the first BT injection (baseline), we collected demographic variables (age and sex) and the following MS-related clinical features: The disease duration (years from reported disease onset to study inclusion), the Expanded Disability Status Scale (EDSS), the current disease-modifying treatment (DMT) for MS (classified into low/medium or high efficacy, in accordance with the Italian regulatory agency), and the total score on the Symbol Digit Modalities Test (SDMT) (evaluating attention and processing speed, and reflecting the overall cognitive status in MS) [20]. The SMDT score was age, sex, and education adjusted and then classified as normal or impaired based on a previous study [21]. 

During the follow-ups, we further collected EDSS progressions (defined as an increase in the EDSS score of ≥1.5 points from an EDSS score of 0.0, ≥1.0 point from an EDSS score of 1.0–5.5, or ≥0.5 point from an EDSS score ≥6.0 [22]), clinical relapses, and DMT changes (as a proxy of clinical activity and/or side effects).

### 5.3. Spasticity and BT Injection Variables

Spasticity was clinically defined as an increase in the velocity-dependent reflexes to phasic stretch, detected and measured at rest [23]. The spasticity evaluation included a separate assessment of the tone in specific muscle groups (e.g., shoulder, elbow, wrist, fingers, hip, leg, knee, and ankle) by using the modified Ashworth score (MAS) (the minimum MAS score for the definition of spasticity was 1); a MAS score of 1+ was coded as 1.5 for statistical purposes. For each patient, at the time of the first BT injection, the highest MAS score among the injected muscles was used for statistical purposes [3]. The concomitant spasticity treatments were also collected, as per the Italian consensus on the treatment of spasticity in MS [24].

We collected the dates of the first and last BT injections, the total number of injections, and the average interval between the injections; BT discontinuation was defined as the absence of further BT injections for at least 6 months after the previous injection. The injection goals were classified using the World Health Organization (WHO) International Classification of Functioning, Disability, and Health (ICF) (http://apps.who.int/classifications/icfbrowser/ accessed on 1 April 2022) [3] into the following categories: posturing/hygiene, mobility, pain, and daily assistance/functioning in daily living activities. We also recorded the BT formulation (abobotulinumtoxin A [Dysport^®^], incobotulinumtoxin A [Xeomin^®^], or onabotulinumtoxin A [Botox^®^]) and dose at the time of the first BT injection and any changes during the follow-ups. In accordance with previous papers on the same topic, for the comparison of patients using different BT formulations, the doses were unified [25]. Since most of the patients had been treated with either incobotulinumtoxin A [Xeomin^®^] or onabotulinumtoxin A [Botox^®^], these doses were left unchanged, while the abobotulinumtoxin A [Dysport^®^] doses were divided by 2.5 to yield comparable unified dose units (uDU). We used ultrasound guidance for the injections and diluted all of the BT formulations to 2 mL, as per our clinical practice [3].

The side effects of the BT injections were also collected; however, we only referred to serious adverse events, which are less likely to be missed in clinical practice (defined as reactions that result in death, are life-threatening, require hospitalization or prolongation of existing hospitalization, result in persistent or significant disability or incapacity, or are a birth defect).

### 5.4. Power Calculation

Considering a normal distribution of the variables to be analyzed, a sample of 108 and 14 patients would be able to achieve 98% power with a 5% α error to detect a 10% change in the risk of discontinuation.

### 5.5. Statistics

The mean (and standard deviation), median (and range), and number (and percent) were calculated for the different study variables, as appropriate. The differences between the MS patients with and without discontinuation of BT were preliminarily evaluated using a *t*-test, a chi-square test, or a Fisher’s exact test, as appropriate.

We used the Cox proportional hazards regression models to assess the hazard of BT discontinuation (a time-dependent variable) in relation to the clinical variables (MS, spasticity, and injections). In particular, we ran three different Cox regression models in turn, including the MS (disease duration, EDSS, and DMT efficacy at the first injection, DMT changes, and impaired SDMTs), spasticity (highest MAS score and concomitant spasticity treatments), and injection variables (BT formulation and BT dose at the first injection, changes in the BT formulation or dose, total number of BT injections, and intervals between the BT injections). The covariates were age and sex. Finally, considering the amount of variables, a parsimonious approach was applied using the Cox regression stepwise model, including all of the above-mentioned variables with a backward selection of *p* = 0.20 as the critical value for entering variables in the model. The proportional hazard assumption was met, as assessed using plots of log (−log survival time) against log survival time and Schoenfeld residuals against survival time; we also used a linear regression of Schoenfeld residuals on time to test for independence between the residuals and the time.

The results were reported as hazard ratios (HR), 95% confidence intervals (95%CI), and *p*-values, as appropriate. The results were considered statistically significant if *p* < 0.05. The statistical analyses were performed using IBM SPSS Statistics 27.0.

## Figures and Tables

**Figure 1 toxins-14-00774-f001:**
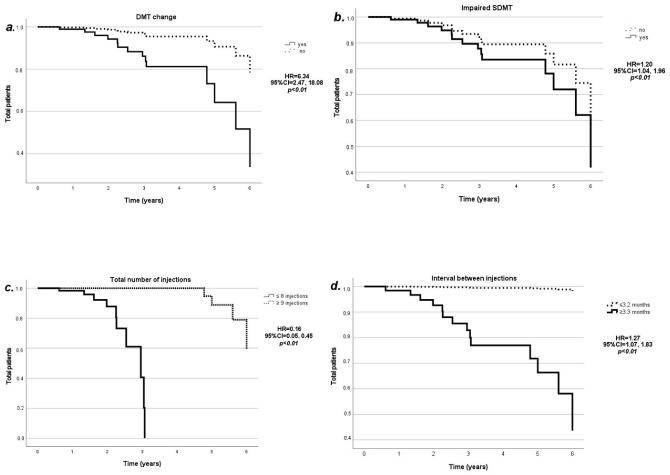
Persistence with the botulinum toxin. Kaplan–Meier curves showing the rate of BT discontinuation in relation to the DMT changes (**a**); impaired SDMT (**b**); total number of injections (**c**); interval between injections (**d**); HR, 95%CI, and *p*-values are shown from the Cox regression models.

**Table 1 toxins-14-00774-t001:** Demographic, MS, spasticity, and injection variables. The *p*-values show the differences between the MS patients continuing or discontinuing the BT injections, using a *t*-test, a chi-square test, or a Fisher’s exact test, as appropriate (* indicates *p* < 0.05).

		BT Continuation (*n* = 108)	BT Discontinuation (*n* = 14)	*p*-Value
Age, years		50.1 ± 9.4	44.0 ± 10.6	0.02 *
Sex, females		46 (42.6%)	6 (42.8%)	0.98
Follow-up duration, years		2.7 ± 1.5	3.0 ± 1.5	0.47
Disease duration, years		14.4 ± 8.3	13.1 ± 9.1	0.58
EDSS		5.8 ± 1.2	5.4 ± 1.3	0.26
DMT	None	9 (8.3%)	0 (0%)	0.30
Low/Medium efficacy		35 (32.4%)	3 (21.4%)	
High efficacy		64 (59.3%)	11 (78.6%)	
DMT change		29 (26.8%)	2 (14.3%)	0.54
SDMT, adjusted score		38.0 ± 11.3	35.4 ± 17.5	0.37
SDMT, impaired		40 (37.0%)	6 (42.8%)	
MAS, highest score		1.8 ± 0.5	1.9 ± 0.5	0.77
Concomitant spasticity treatments		48 (44.4%)	6 (42.8%)	0.91
BT formulation	Botox	39 (36.2%)	7 (50.0%)	0.56
Dysport		45 (41.6%)	4 (28.6%)	
Xeomin		24 (22.2%)	3 (21.4%)	
BT dose, uDU		263.4 ± 157.0	225.0 ± 131.19	0.38
BT changes		28 (25.9%)	4 (28.6%)	0.83
Total number of BT injections		10.3 ± 5.5	8.1 ± 5.6	0.17
Interval between BT injections, months		3.1 ± 0.4	4.9 ± 1.3	<0.01 *

## Data Availability

Data is available upon reasonable request to the corresponding author.

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
