# Peer review of "Persistence with Botulinum Toxin Treatment for Spasticity Symptoms in Multiple Sclerosis"

_toxins, 2022, doi:10.3390/toxins14110774_

Round 1

Reviewer 1 Report

This is a well designed, well done study. However, there is limited insight w respect to the drivers of BT discontinuation that should be further inspected, for example, new treatments / new modalities, lost to follow-up, further worsening of MS, cognitive decline

Author Response

  • This is a well designed, well done study. However, there is limited insight with respect to the drivers of BT discontinuation that should be further inspected, for example, new treatments / new modalities, lost to follow-up, further worsening of MS, cognitive decline.

We thank the reviewer for his/her feedback. We have now revised the following sentences in the discussion: “We did not collect reasons for discontinuation, that can include logistics, switch to other spasticity treatments, poor efficacy and/or tolerability. However, looking at our results, disease worsening in terms of cognitive impairment and DMT failure are the most likely responsible for botulinum toxin discontinuation”. We also added appropriate references.

Reviewer 2 Report

Interesting study about factors related to the persistence to botulinum toxin treatment for spasticity in people with multiple sclerosis. However, there are several points that could be improved: 

- Introduction: 

Why is this study important? What is the gap it fills? What do previous studies focusing on short-term effects show? 

The aim of the research should be clearly stated, not only the hypothesis. 

- Methods: 

Has the required sample size been calculated? If not, it should be indicated and included as a limitation of the research.  

The registration number of the ethics committee that evaluated the study have to be included. 

It seems that the type of multiple sclerosis (relapsing-remitting, primary progressive, etc.) has not been collected. This could be an important factor to include in the analysis. 

It seems that concomitant spasticity treatment has been collected, but it is not stated what these treatments are. It would be interesting to show them. In addition, they may be factors to consider in the analysis. 

MAS has been used to assess spasticity; however, has the 1+ value been included? In Table 1 it appears that MAS is shown as a quantitative variable, with no indication of how the 1+ value was coded. 

The target of botulinum toxin is shown (according to WHO), but nothing is showed about the target muscles. This could be interesting in the analysis.

- Results: 

The number of people in the BT discontinuation group is only 14. Is the sample size right for the objectives set? In addition, there are differences between groups in age, which should be considered in the interpretation of the results. 

- Discussion: 

Other research limitations should be considered, as previously indicated (sample size, lack of information on the type of SM, etc.). 

- Conclusion: 

The conclusions should respond more clearly to the objectives of the study. The last one is more appropriate as part of the discussion. 

Author Response

  • Interesting study about factors related to the persistence to botulinum toxin treatment for spasticity in people with multiple sclerosis. However, there are several points that could be improved:

We thank the reviewer for his/her feedback. Please, find below point-by-point response to your comments.

  • Introduction: Why is this study important? What is the gap it fills? What do previous studies focusing on short-term effects show? The aim of the research should be clearly stated, not only the hypothesis.

As suggested, we have added the following sentences to the introduction: “Clinical trials are generally run in a limited time frame (i.e. response at 4-12 weeks), and include a variety of neurological conditions, thus limiting the analysis on disease-specific features. Similarly, real-world studies have focused on short-term improvements following BT injection (e.g., up to 12 months), while long-term use has been poorly investigated” and “we… aim to evaluate the risk of discontinuation in relation to MS clinical, spasticity and injection-related features. We hypothesize that there are individual factors increasing the risk of discontinuation,  that should therefore be considered in clinical practice within customized treatment decisions”.

  • Methods: Has the required sample size been calculated? If not, it should be indicated and included as a limitation of the research.

We thank the reviewer for suggesting this, and have now added a power calculation: “Considering a normal distribution of variables to be analysed, a sample of 108 and 14 patients would be able to achieve 98% power, with 5% α error, to detect 10% change in the risk of discontinuation”.

  • Methods: The registration number of the ethics committee that evaluated the study have to be included.

We have now added the registration number.

  • Methods: It seems that the type of multiple sclerosis (relapsing-remitting, primary progressive, etc.) has not been collected. This could be an important factor to include in the analysis.

We have now acknowledged this in the limitations’ section of the manuscript.

  • Methods: It seems that concomitant spasticity treatment has been collected, but it is not stated what these treatments are. It would be interesting to show them. In addition, they may be factors to consider in the analysis.

Indeed, we have included concomitant spasticity treatments in our statistical models, but did not find significant associations. We have now included detailed spasticity treatments in the results.

  • Methods: MAS has been used to assess spasticity; however, has the 1+ value been included? In Table 1 it appears that MAS is shown as a quantitative variable, with no indication of how the 1+ value was coded.

We have now added in the methods that MAS 1+ was coded as 1.5, as from our previous study.

  • Methods: The target of botulinum toxin is shown (according to WHO), but nothing is showed about the target muscles. This could be interesting in the analysis.

Unfortunately, we did not include target muscles in the analyses, that however would have been difficult to analyse due to sample size constraints. We have added the following sentence to the limitations’ paragraph of the discussion: “We did not analyse target muscles but preferred injection goals due to sample sites constrains and to the willingness to evaluate actual benefit in daily life activities”.

  • Results: The number of people in the BT discontinuation group is only 14. Is the sample size right for the objectives set? In addition, there are differences between groups in age, which should be considered in the interpretation of the results.

We have now included a power analysis, as from your previous comment. In our statistical models age was not associated with the risk of discontinuation.

  • Discussion: Other research limitations should be considered, as previously indicated (sample size, lack of information on the type of SM, etc.).

As suggested, we have now revised the limitations’ paragraph of the discussion.

  • Conclusion: The conclusions should respond more clearly to the objectives of the study. The last one is more appropriate as part of the discussion.

We have now revised the conclusion as follows: “Overall, MS and BT specialists should share an appropriate treatment plan, also with patients with MS, and including a combination of spasticity, BT injection and MS variables, to foster long term persistence”.

Reviewer 3 Report

Dear authors, I had the pleasure of revising the article entitled "Persistence to botulinum toxin treatment for spasticity symptoms in multiple sclerosis".

The article shows a complete study regarding an important clinical pathology, being also a topic of particular interest to the journal.

The methodological part is well described, following the indications of the authors. The table is clear to read, include all critical information and are complete.

The discussion is adequate, as are the conclusions, maybe the authorship can complete the text for conclusions, because the informations are very valuables.

Apart from these comments, the structure of the work appears worthy of publication.

Author Response

  • Dear authors, I had the pleasure of revising the article entitled "Persistence to botulinum toxin treatment for spasticity symptoms in multiple sclerosis". The article shows a complete study regarding an important clinical pathology, being also a topic of particular interest to the journal. The methodological part is well described, following the indications of the authors. The table is clear to read, include all critical information and are complete.

We thank the reviewer for his/her feedback.

  • The discussion is adequate, as are the conclusions, maybe the authorship can complete the text for conclusions, because the information are very valuables. Apart from these comments, the structure of the work appears worthy of publication.

We have now improved the conclusions’ paragraphs as suggested.

Round 2

Reviewer 2 Report

The comments have been answered satisfactorily. Thank you very much.